# Micro-Encapsulated Microalgae Oil Supplementation Has No Systematic Effect on the Odor of Vanilla Shake-Test of an Electronic Nose

**DOI:** 10.3390/foods11213452

**Published:** 2022-10-31

**Authors:** Haruna Gado Yakubu, Omeralfaroug Ali, Imre Ilyés, Dorottya Vigyázó, Brigitta Bóta, George Bazar, Tamás Tóth, András Szabó

**Affiliations:** 1Agribiotechnology and Precision Breeding for Food Security National Laboratory, Institute of Physiology and Nutrition, Department of Physiology and Animal Health, Hungarian University of Agriculture and Life Sciences, Guba Sándor utca 40, 7400 Kaposvár, Hungary; 2HuBakers Trade Ltd., ÁTI-Sziget Ipari Park 11, 2310 Szigetszentmiklós, Hungary; 3ELKH-MATE Mycotoxins in the Food Chain Research Group, Kaposvár Campus, Hungarian University of Agriculture and Life Sciences, Guba Sándor utca 40, 7400 Kaposvár, Hungary; 4ADEXGO Kft., Lapostelki utca 13, 8230 Balatonfüred, Hungary

**Keywords:** fortification, odor profiling, machine olfaction, docosahexaenoic acid, food enrichment, functional food

## Abstract

In this study, we aimed to carry out the efficient fortification of vanilla milkshakes with micro-encapsulated microalgae oil (brand: S17-P100) without distorting the product’s odor. A 10-step oil-enrichment protocol was developed using an inclusion rate of 0.2 to 2 *w*/*w*%. Fatty acid (FA) profile analysis was performed using methyl esters with the GC-MS technique, and the recovery of docosahexaenoic acid (C22:6 n3, DHA) was robust (r = 0.97, *p* < 0.001). The enrichment process increased the DHA level to 412 mg/100 g. Based on this finding, a flash-GC-based electronic nose (e-nose) was used to describe the product’s odor. Applying principal component (PC) analysis to the acquired sensor data revealed that for the first four PCs, only PC3 (6.5%) showed a difference between the control and the supplemented products. However, no systematic pattern of odor profiles corresponding to the percentages of supplementation was observed within the PC planes. Similarly, when discriminant factor analysis (DFA) was applied, though a classification of the control and supplemented products, we obtained a validation score of 98%, and the classification pattern of the odor profiles did not follow a systematic format. Again, when a more targeted approach such as the partial least square regression (PLSR) was used on the most dominant sensors, a weak relationship (R^2^ = 0.50) was observed, indicating that there was no linear combination of the qualitative sensors’ signals that could accurately describe the supplemented concentration variation. It can therefore be inferred that no detectable off-odor was present as a side effect of the increase in the oil concentration. Some volatile compounds of importance in regard to the odor, such as ethylacetate, ethyl-isobutarate, pentanal and pentyl butanoate, were found in the supplemented product. Although the presence of yeasts and molds was excluded from the product, ethanol was detected in all samples, but with an intensity that was insufficient to cause an off-odor.

## 1. Introduction

Micro-encapsulation is a widely used and accepted form of flavor and aroma preservation in the food industry, and it has been an important technique for a long time. Micro-encapsulation is not only successful in preserving or masking flavor and aroma compounds in foods; it has also been found to enhance the thermal and oxidative stability of aromatic compounds in foods [1].

The application of micro-encapsulation in food systems has also been carried out with the aim of overcoming the challenges of high volatility, or to control the fast release and to improve the poor bioavailability of bioactive compounds [1,2,3]. Many techniques, such as spray-drying (SD), freeze-drying (FD), coacervation, spray granulation (SG), emulsification, the use of supercritical fluids (SCFs), and electrospraying [4,5,6,7,8], have been reported as successful encapsulation methods of docosahexaenoic acid (C22:6n3, DHA) in foods. The success of DHA encapsulation may also depend on the composition of the encapsulation wall material [9,10]. Due to the rapid growth of the food industry sector, there have been possibilities to improve the nutritional quality of products through the development of so-called “functional foods” [11,12]. Microalgae biomass and oil have been widely incorporated into foods and beverages [13] during the past decade. Microalgae oil has been successfully incorporated into foods such as ice cream, milk drinks [14,15], sausages [16], yoghurt [17], and cheeses [18,19], increasing the omega 3 fatty acid (n3 FA) content of these products without altering their odor.

The addition of microalgae oil to a vanilla milkshake can improve its fatty acid (FA) composition through increasing the bioactive n3 FA proportions, such as eicosapentaenoic acid (C20:5n3, EPA) and DHA. These FAs have been documented to reduce the risk of coronary heart disease and inflammatory disorders [20,21].

Ideally, fish and microalgae oils, which are widely known to be rich in DHA, may also lead to an unpleasant odor and flavor, which is a product of their poor oxidative stability [22], and may affect or limit their use as nutraceuticals or as functional food agents. Interestingly, some methods have been applied over the years to stabilize highly polyunsaturated oils in foods, such as the addition of antioxidants to the bulk oil. However, this does not allow the successful removal of all the unpleasant flavors or odors [6,9], and encapsulation has been very widely used to achieve this purpose.

The sensory characteristics of foods enriched with DHA, such as their odor and taste, can vary according to the levels of supplementation. In the case of odor, the electronic nose (e-nose) is currently one of the most widely used technologies for rapid aroma profiling of foods and has proven to be an efficient sensory technology [23,24]. The e-nose, in some instances, is used as the best complementary technique to validate conventional or traditional methods of odor profiling and can even substitute those when technical conditions are required [25]. As a rapid analytical system or method, the e-nose is made up of three major parts: the sample delivery, detection, and computing systems [26].

To date, numerous technological solutions have been made available, using various materials, such as sensor arrays with metal oxide semiconductor sensors (MOS), the metal-oxide-semiconductor field-effect transistor (MOSFET), conducting polymer composites and intrinsically conducting polymers [27,28,29], and the gas chromatography (GC)-based electronic nose. Hence, applying chemometrics to e-nose sensory data helps to discriminate between various identified volatile compounds [30].

The milkshake is a very common dairy product that is consumed by both young people and adults due to its easy/rapid method of preparation and its desirable flavors. On the other hand, S17-P100 is a micro-encapsulated microalgae oil product that is known to be rich in EPA and DHA, which is also commonly available on the European market. Though the fortification of food products using microalgae oil is not new, limited information is available concerning its use in milkshake fortification and, more importantly, concerning how the e-nose could be applied to verify the success of this fortification process and to ensure the final product’s quality.

In this study we therefore aimed to ascertain the efficacy of the addition of the n3 FA fortification brand (micro-encapsulated microalgae oil) in fulfilling the dietary recommendations for n3-FA-enriched food products, and to test the application of the e-nose in profiling the odor of n3-FA-enriched vanilla milkshakes. The basic hypothesis to be tested was whether or not the characteristically oxidative or fishy odor associated with micro-encapsulated microalgae oil had a detectable systematic distorting effect on the odor profile of the fortified vanilla shake product.

## 2. Materials and Methods

### 2.1. Experimental Shake Powder

The vanilla shake powder used in this study was a product available on the market, of which the ingredients are given in Table 1.

### 2.2. Determination of the Product’s Fatty Acid Profile

Samples (shake powder, FA additive, and complemented shake powder) were homogenized (IKA T25 Digital Ultra Turrax, Staufen, Germany) in a 20-fold volume of chloroform:methanol (2:1 *v*:*v*) and total lipid content was extracted according to the method of Folch et al. [31]. Solvents were ultrapure-grade (Carl Roth, Karlsruhe, Germany) and 0.01% *w*/*v* butylated hydroxytoluene was added to prevent FA oxidation. Directly to the raw, dry sample, C19:0 internal standard was added (Merck cat. No.: 72332). The internal standard used was a solution of 1 mg/mL in chloroform:methanol (2:1 *v*:*v*). The total amount added was ca. 1/20 mass of the extracted fat, i.e., to 1 g raw sample (ca. 100 mg crude extract), 5 mg C19:0 was added.

The total lipid extract (also including the internal standard) was dried fully on a rotary evaporator under nitrogen stream and was trans-methylated via the acid-catalyzed method [32], using H_2_SO_4_ (1 *v*/*v*%) in methanol as a methyl donor, and toluene was used as a solvent. For the quantitative analysis, C19:0 methyl ester standard calibration was used at 6 points (Merck cat. No.: 74208) to assess the detector response, and the concentration of analyte in the calibration was between 5 and 500 μg/mL. The correlation coefficient was not less than 0.999, proving the linearity of the analysis. Fatty acid methyl-esters were extracted into ultrapure n-hexane for gas chromatography. This was performed on a Shimadzu GCMS-QP2010 apparatus (AOC 20i automatic injector), equipped with a Phenomenex Zebron ZB-WAX Capillary GC column (30 m × 0.25 mm ID, 0.25 μm film, Phenomenex Inc., Torrance, CA, USA). The characteristic operating conditions were: injector temperature: 270 °C, detector temperature: 300 °C, helium flow: 28 cm/s. The oven temperature was graded from 80 °C to 205 °C: 2.5 °C/min, 5 min at 205 °C, from 205 °C to 250 °C 10 °C/min and 5 min at 210 °C. FA results are expressed as mg/g of raw sample mass, as well as a weight% of the total FAs. All samples were analyzed in duplicate, and results are means of 2 analyses. The limit of detection was determined as three times the signal-to-noise ratio (3S/N), whereas the limit of quantification was 10S/N. The range of the LOD was between 0.1 and 0.5 μg/mL for the FAs (C4:0 to C24:0).

### 2.3. Omega-3 Fatty Acid Enrichment Protocol

To perform a graded n3 FA fortification protocol (primarily docosahexaenoic acid enrichment), a micro-encapsulated marketed algae-oil-based product was chosen, with the brand name S17-P100 (Life’s DHA, DSM Nutritional Products Inc., Heerlen, The Netherlands).

The original shake mixture and the food additive product were subjected to FA analysis, as shown in Table 2 and Table 3, respectively. The graded enrichment protocol for the vanilla shake powder with the FA based product is shown in Table 4.

### 2.4. Aroma Analysis with the Electronic Nose

The odors of vanilla shake powders were measured in four replicates, 10 days after the enrichment protocol presented in Table 4. Before the odor measurement, three 1 g aliquots of each sample were placed into three 20 mL headspace vials, sealed with a magnetic cap and an UltraCleanTM polytetrafluoroethylene/silicone septum (Supelco, Inc., Merck KGaA, Darmstadt, Germany). The e-nose measurement was performed with a Heracles Neo 300 ultra-fast GC analyzer (Alpha MOS, Toulouse, France), equipped with a PAL-RSI autosampler unit (CTC Analytics AG, Zwingen, Switzerland) for standard handling of the samples. We generated the headspace and injected the headspace into the Heracles analyzer unit, including an odor concentrator trap and two metal capillary columns: (1) Restek MXT-5: length: 10 m, ID: 0.18 mm, thickness: 0.40 μm, low-polarity stationary phase composed of Crossbond 5% diphenyl/95% dimethyl polysiloxane (Restek, Co., Bellefonte, PA, USA); (2) Restek MXT-1701: length: 10 m, ID: 0.18 mm, thickness: 0.40 μm, mid-polarity stationary phase composed of Crossbond 14% cyanopropylphenyl/86% dimethyl polysiloxane (Restek, Co., Bellefonte, PA, USA). The volatile compounds were separated by both columns simultaneously and detected using two flame ionization detectors (FIDs). The autosampler and analyzer were operated using AlphaSoft ver. 16 (Alpha MOS, Toulouse, France). The same software was used for the data acquisition and transformation. The retention times of the volatiles were recorded at both FIDs, followed by a conversion to retention indices (RIs). The Kovats RIs relate the retention times of the detected volatile molecules of a sample to the retention times of n-alkanes under the same conditions [33].

The RIs characterize the volatile compounds on the specific columns and can be assigned to specific aromas recorded in AroChemBase v7 in the AlphaSoft software. Throughout this manuscript, “1-A” is used as an identifier after RI to refer to column MXT-5, and “2-A” is used to refer to column MXT-1701. Before the analysis, a method was developed with the following parameters of the PAL-RSI Autosampler and Heracles GC analyzer: (1) autosampler: incubation at 80 °C for 10 min with 500 rpm agitation to generate headspace, 5 mL of headspace injected into the Heracles analyzer, flushing time between injections: 90 s; (2) analyzer: carrier gas: hydrogen, the flow of carrier gas: 30 mL/min, trapping temperature: 60 °C, initial oven temperature: 50 °C, the endpoint of oven temperature: 250 °C, heating rate: 2 °C/s, acquisition duration: 110 s, acquisition period: 0.01 s, injection speed: 125 μL/s, cleaning phase: 8 min.

### 2.5. Microbiological Testing

To elucidate the possible origins of ethanol in the sample, the enumeration of viable osmophilic yeasts and xerophilic molds was performed with conventional agar plate testing [34] using Dichloran-Glycerol (DG18) agar.

### 2.6. Statistical Evaluation

The recovery, i.e., the relationship between the calculated and measured amount of DHA, was implemented as a linear relationship between the two variables. The linear regression model was calculated with QtiPlot (version 1.0.0., 2020).

The multivariate data of the e-nose measurements describing the odor profiles of the S17-P100-enriched vanilla shake samples were analyzed using AlphaSoft (ver. 16) software. The chromatograms were transformed into a series of variables called sensors based on the identified chromatogram peaks [35]. The name of a sensor originating from the location of the peak within the chromatogram and was identical to the respective RI. The sensor intensity was calculated based on the area under the respective chromatogram peaks.

Principal component analysis (PCA) [36] was performed using the sensor data to describe the unsupervised clustering of the samples within the multidimensional space defined by the sensor variables. Supervised classification models were built using discriminant factor analysis (DFA) [36] to identify the predefined groups of samples based on their odor signals. Partial least-squares regression (PLSR) [36] was used to fit calibration models describing the relationship between the odor signals and the concentration of the S17-P100 food additive.

The accuracy of the DFA and PLSR models was tested with leave-one-out cross-validation, in which a single record was left out of the modeling process and was used for testing by predicting its qualitative or quantitative properties; this process was repeated iteratively until all samples had been used for validation once [36]. The sensor selection function of AlphaSoft was used to identify the most distinctive variables during the qualitative and quantitative analyses. In addition, DFA and PLSR calculations based on the selected sensors were performed. The volatile compounds described by the selected sensors were identified using the AroChemBase database.

## 3. Results

### 3.1. Fatty Acid Profile of the Enriched Shake

The enriched vanilla shake powder demonstrated a gradual increase in the proportions (and concentrations) of the medium- and long-chain FAs (FAs with carbon chains > 16) that were abundant in the FA additive. Consequently, this was visible in the overall increasing level of polyunsaturation, as well as that of monounsaturation, whereas the n6/n3 FA proportion decreased. The detailed FA profile of the enriched vanilla shake is provided in Table 5.

### 3.2. Fatty Acid Recovery from the Enriched Shake

With the S17-P100-enriched vanilla shake powder, FA analysis was performed. To test the mixing accuracy, we analyzed the relationship between the estimated and the recovered amount of the largest additive component, docosahexaenoic acid (DHA, C22:6n3) (Figure 1).

The R^2^ value was found to be relatively robust, but the intercept was found to be significantly different from zero (*p* < 0.001).

### 3.3. Microbiology

The counts of osmophilic yeasts and xerophilic molds both remained below 100 CFU/g of product, i.e., below the limit of detection.

### 3.4. Electronic Nose

The chromatograms recorded on the two GC columns during the analysis of the headspace of one vanilla shake sample are shown in Figure 2. Based on the retention indices, the main identified volatile compounds were ethanol (463-1-A; 560-2-A), propylene glycol (723-1-A; 950-2-A), limonene (1043-1-A; 1070-2-A), and vanillin (1437-1-A; 1706-2-A).

The main volatiles mentioned above were responsible for the dominant smell of the product, but a larger fraction of the variation in the multicomponent odor fingerprint data was influenced by the smaller chromatogram peaks. To describe the multivariate odor patterns, principal component analysis (PCA) was performed upon all the sensor signals that were derived from the chromatograms (as described in Section 2). The first (PC1) and second (PC2) principal components explained 72.2% and 19.4% of the e-nose data variation, respectively, without any systematic (increased or decreased) pattern of odor differences between the supplemented samples and the control.

The odor difference between the control and supplemented samples appeared only in the 3rd principal component (PC3) (a higher PC) which described 6.5% of the total sensor signal variation (Figure 3). Even though this latent variable described the difference in odor between the control and supplemented samples, it had no relation with the level of S17-P100 supplementation, as no systematic arrangement among treatments could be identified. Though the variation in the odor profile may not have been negligible, this component did not describe the dominant odor pattern of the supplemented milkshakes that may affect consumer choice.

The supervised classification of samples was performed with discriminant factor analysis (DFA), using the data from all sensors. The samples containing different concentrations of the supplementation could be identified with 98% accuracy during the cross-validation (Figure 4), but the arrangement of the groups within the plane of discriminant factors did not follow the concentration level of the supplementation (a similar observation to that of the unsupervised classification (PCA)). This demonstrates that the DFA model could find odor (or odor intensity) differences between sample groups, but these differences were not influenced by the level of supplementation due to the lack of a systematic effect on odor intensities in the DFA plane.

On the other hand, when the relation between the sensor data and the supplementation level was described with partial least-squares regression (PLSR), an accurate (R^2^ = 0.833) calibration model was built (Figure 5), which indicated the possibility of finding supplementation-dependent odor variations. To further investigate this seeming possibility, the specific sensors were selected that contributed the most to the successful classification of samples with different levels of supplementation. Figure 6 shows the mean sensor signals for each concentration group. It was noticeable that the supplemented samples had higher intensities at 547-1-A, 612-1-A, 756-1-A, 802-1-A, 996-1-A, and 778-2-A than the control samples. The possible volatile compounds identified for these retention indices in the AroChemBase are shown in Table 6. PLSR calibration and sensor selection based on PLSR calibration are targeted approaches to finding specific odor differences focusing on predefined parameters or constituents; thus, finding the relevant odor variations is possible even if the causative odorants are very weak. However, an approach that is non-targeted, such as PCA, is relevant when identifying general odor patterns, focusing on the major odor variations, which in this paper revealed non-dependent odor differences in relation to the increased levels of DHA supplementation.

In a supervised DFA (Figure 7) classification of samples with different S17-P100 contents based on the data from the selected qualitative sensors (targeted modeling), the model was able to sort 98% of the samples into the correct classes. Again, no concentration-dependent arrangement of the classes could be recognized in this model.

Although the data from the selected qualitative sensors (targeted modeling) held enough information to differentiate the samples based on the DHA inclusion levels in the S17-P100 supplementation, this differentiation was not based on volatiles that had a direct relationship with the supplemented concentration. This is demonstrated in Figure 8, which shows the results of the PLSR calibration on the supplementation level using the data from the selected sensor set. The weak results (R^2^ = 0.509) indicate that there was no linear combination of the qualitative sensors’ signals that could accurately describe the variation in supplemented concentrations.

In a final attempt, sensors which gave an optimal linear combination for the PLSR calibration of the concentration of S17-P100 supplementation were selected. The largest variation and the variation that was proportional to the concentration of supplementation was found in sensor 756-1-A, referring to the retention index of ethyl isobutyrate (or that of pyrrole). The good results (R^2^ = 0.813) of the PLSR calibration shown in Figure 9 demonstrate that the linear combinations of the limited number of selected quantitative sensors held enough information to describe the concentration-dependent odor variations caused by S17-P100 supplementation. The calibration fitted onto the data from these few sensors was almost as good as that obtained with all the sensors (Figure 5). However, as explained earlier, this targeted method of classification or modeling using specific or selected signals is not good enough to correctly describe the major odor differences.

## 4. Discussion

### 4.1. Original Fatty Acid Profile of the Shake Powder

The original shake powder’s FA profile typically resembled that of coconut fat, with a slight modification due to the effect of milk whey. The FA profile of the un-complemented (i.e., original) product definitely indicated the dominance of saturated FAs (~81%), with lauric (C12:0, ~30%), palmitic (C16:0, ~18%), and myristic (C14:0, ~16%) acids as the main components. The major monoenoic FA was oleic acid (C18:1 n9), with a share of 13%, and the most abundant polyunsaturated one was linolenic acid (C18:2n6, slightly over 4%) (Table 3). The n3 FAs were present only minimally, with α-linolenic acid (C18:3n3) providing 0.23% and eicosapentaenoic acid (C20:5 n3) accounting for 0.04%. The n3 FAs of the C22 chain length were absent from the original sample.

The dominant FA source ingredient was coconut oil. Coconut oil is available in three forms: refined oil, copra oil, and virgin coconut oil. The FA profiles of the different types were practically uniform [37]. The form used here was powdered coconut fat. According to [37,38], coconut oil is the highest natural source of lauric acid, and possesses high proportions of caprylic, capric and lauric acids; these were detected in our sample as well, in proportions of ~3%, ~4.5%, and ~30%, respectively.

In summary, we implemented the n3 FA enrichment of a food product that had a very strongly saturated FA profile and of which the medium-chain length acids were already quite abundant.

### 4.2. Meeting the Dietetic Recommendations Regarding Omega-3 Fatty Acids for Humans

Vanilla aroma shakes are not easy to classify within the wide varieties of milk products. This product is similar to most infant formula products (composed of 70% milk whey) and it is prepared in a ready-to-drink form with the addition of water. The basic rationale behind the addition of n3 long chain FAs from microalgae to a product that is originally poor in them (Table 3) is absolutely clear.

The globally recommended levels for EPA (C20:5n3) and DHA (C22:6n3) are generally handled together and are mostly specific for gender, health status, and age. In 2002, the WHO recommended a total daily amount of n3 FA of 1–2% of the total energy intake for adults, whereas for EPA + DHA, the recommended range is from 100–150 mg/day (2–4 years) and up to 300 mg/day (pregnant or lactating women) [39]. The European Food Safety Authority also recommends 250 mg day^−1^ DHA + EPA for adults [40]. The highest global recommendation is at least 500 mg EPA + DHA/day/capita [41]. The US [42] and NATO recommendations [43] are quite similar to that of the FAO [39], at 300–400 mg EPA + DHA/day, and slightly exceed the pregnant and lactating women’s recommendations (Koletzko et al. suggested 200 mg EPA + DHA) [44]. Thus, in this study we aimed at a range covering and possibly slightly exceeding most of the above-cited doses by using S17-P100 levels above 100 mg/10 g milkshake powder (with a theoretical maximum of 428 mg DHA and 16 mg EPA in 100 g).

In summary, in practice the recommended levels were nearly reached in this study, and we may add that the relatively high level of micro-encapsulated microalgae oil added to this product (from 0.2% to 2%, i.e., an increase of one order of magnitude) has been supposed to induce an off-odor corresponding systematically to the increase in the added levels of DHA.

### 4.3. The Possibly Negative Side-Effects of Omega-3 Enrichment

To fulfill consumer demands, microalgae n3 polyunsaturated FAs are present in an increasing number of newly marketed foods [45], including milk products and milky emulsions [46]. With a high degree of polyunsaturation, the *differentia specifica* contribution to the high nutritional/biological value of the fats of these food items makes them prone to lipid oxidation [47]. It is possible that the emerging oxidation products may not only have negative health effects (for a review, see [48]), but their positive nutritional effects may also be compromised by undesirable aromatic properties and thus reduced consumer acceptance [49]. Even if lipid peroxidation can be mostly inhibited with effective methods such as the application of one or more antioxidants [45], the enrichment of emulsified foods with n3 FAs gives rise to undesirable and rancid off-flavors, in most cases making some kind of masking approach necessary [50]. Herein, the e-nose analysis approach was performed to identify possible variations in the odor patterns related to the n3 FA enrichment levels. Based on the data obtained from e-nose sensors (for a detailed discussion, see Section 4.4), we speculate that the negative oxidative effect of n3 polyunsaturated fatty acids was minimal—at least, it did not compromise the odor patterns.

### 4.4. E-Nose Odor Profiling

With the application of the electronic nose system, our aim was to detect how the odor pattern was modified when different levels of micro-encapsulation of DHA containing microalgae oil (SP17-100) were added to the vanilla milkshakes.

When a more targeted approach, such as PLSR calibration using selected sensors, was used to analyze S17-P100 supplementation, a successful calibration could be achieved, which indicated that an odor pattern did exist that was related with the concentration of the additive, i.e., the concentration of S17-P100 had an impact on the odor. However, since the classification models that identified the groups of the different supplementation levels did not identify the concentration-dependent odor patterns, it can be stated that the major odor patterns were independent of S17-P100 supplementation, and these major odor patterns provided the possibility of identifying the sample groups. The minor odor patterns which were dependent on the S17-P100 supplementation concentration could be traced when specifically targeted, but they were generally hidden by the major (independent) odor patterns. Few studies have examined the application of the electronic nose to the odor profiles of milkshakes enriched with n3 FAs. However, some valuable information has been obtained on other related products of functional food value. For instance, our findings are consistent with those of [51], who reported that the addition of *Pavlova lutheri* (an algae) lipid extract to yoghurt increased the n3 FA content without altering the odor 28 days after manufacturing. However, contrary results were obtained in a study in which the odor stability of milk-based infant formula fortified with PUFAs (ARA (C20:4 n6) and DHA (C22:6 n3)) was measured after 30, 60, and 90 days of production [52]. The sixty- and ninety-day odor profiles significantly differed from that obtained at thirty days.

Though the odor profile of the supplemented product did not significantly differ from the controls, there were some volatile compounds of importance in regard to odor that were identified based on retention indices. As shown in Table 6, some of the associated volatile compounds of importance with regard to odor that were identified in the enriched samples were 1-propanol and 2-propanol (isomeric forms of propanol (an alcohol) molecule with a strong odor), ethyl acetate or ethyl ethanoate (an ester formed between acetic acid and ethanol), benzyl butanoate and ethyl hexanoate (a fatty acid ethyl ester) with fruit-like odors [53,54,55], 1,4-cineole (an oxabicycloalkane with a minty lime-oil odor [56]), and pentanal (a saturated fatty aldehyde with an acrid odor [55]). In other studies in which other n3 sources were used for the enrichment process, other volatile compounds of importance with regard to odor were found. In a study in which the volatile compounds in milk enriched with 5% cod oil were measured, the major compounds of importance with regard to odor were trans-2-hexenal and cis-4-heptenal [57], which have a characteristic green leafy or oily odor. Hen et al. attempted to predict the fishy off-odor and identify the compounds of importance with regard to odor in dairy powders, reporting that volatile ketones, aldehydes and furans were the compounds that contributed the most to the identified fishy off-odor [58]. It has been widely reported that the main chemical process underlying the formation of an off-odor is oxidation/the development of rancidity. With respect to fish oil, the oxidation of fish oil is enhanced or increased when it is added to food products. For the quality control of omega-3 PUFA-fortified foods, efficient methods of oxidation assessment should be used, starting from the raw material and continuing during the processing, storage, and distribution of the products [59], and also employing preservation methods such as encapsulation, which may help to mask or hide off-odors. It is also important to always employ rapid quality-checks to detect possible volatile compounds of importance in relation to oxidation or off-odors [60], and the electronic nose could be one of the precise rapid analytical tools to be used for this purpose.

## 5. Conclusions

Adding micro-encapsulated microalgae oil into milkshakes enriched the concentration of n3 FAs, notably DHA (concentration > 400 mg/g), which may contribute to cardiovascular health benefits. When the e-nose was applied to detect the influence of this supplementation process on odor, the odor of the supplemented samples did not show any systematic concentration-dependent pattern as a result of supplementation. Since this observation was evident in most of the models generated, it can be inferred that no major off-odor occurred as a result of the increased level of supplementation. If any major off-odor was present, it must have been manifested in an increased odor pattern with the level of supplementation. However, when targeted signals were selected and used in the modeling process (PLSR), a possible concentration-dependent odor could be calibrated, but this may not represent the major odorants in the product. The presented e-nose approach has high potential in the evaluation of value-added functional foods fortified with health-promoting additives, and may serve as a supportive tool for the development of healthier foods with well-described sensory parameters. The demonstrated e-nose results provided evidence on the major odor profiles of the investigated products. However, human perception may be greatly influenced by several factors, such as but not limited to combinations of smells and mixed sensations of odor, taste and texture. Therefore, further studies, also including human sensory panels, are necessary in order to establish the effect of micro-encapsulated oil on other sensory qualities, such as flavor, color, and taste, to demonstrate consumer acceptance.

## Figures and Tables

**Figure 1 foods-11-03452-f001:**
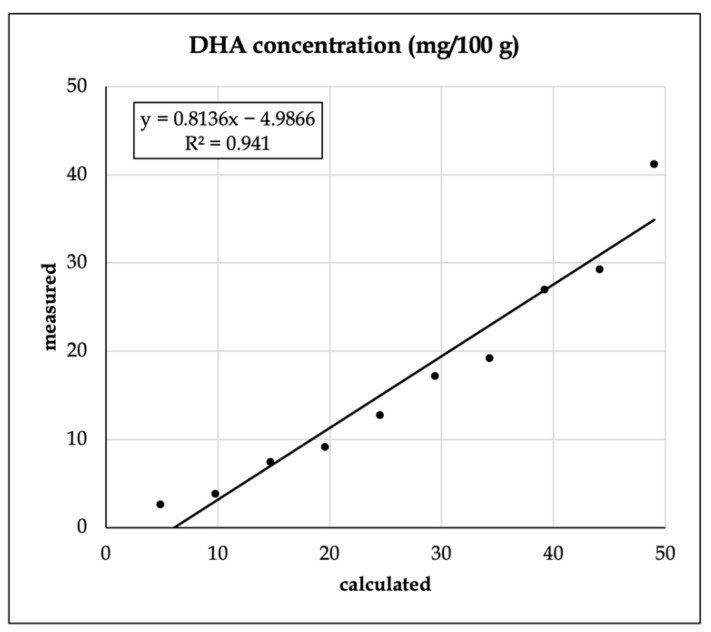
Recovery of docosahexaenoic acid (C22:6 n3, DHA) from the enriched vanilla shake product.

**Figure 2 foods-11-03452-f002:**
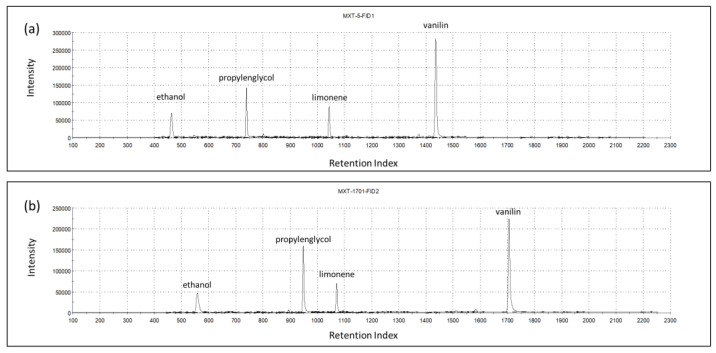
Chromatograms of columns MXT-5 (**a**) and MXT-1701 (**b**) for one vanilla shake sample, indicating the most prominent volatiles.

**Figure 3 foods-11-03452-f003:**
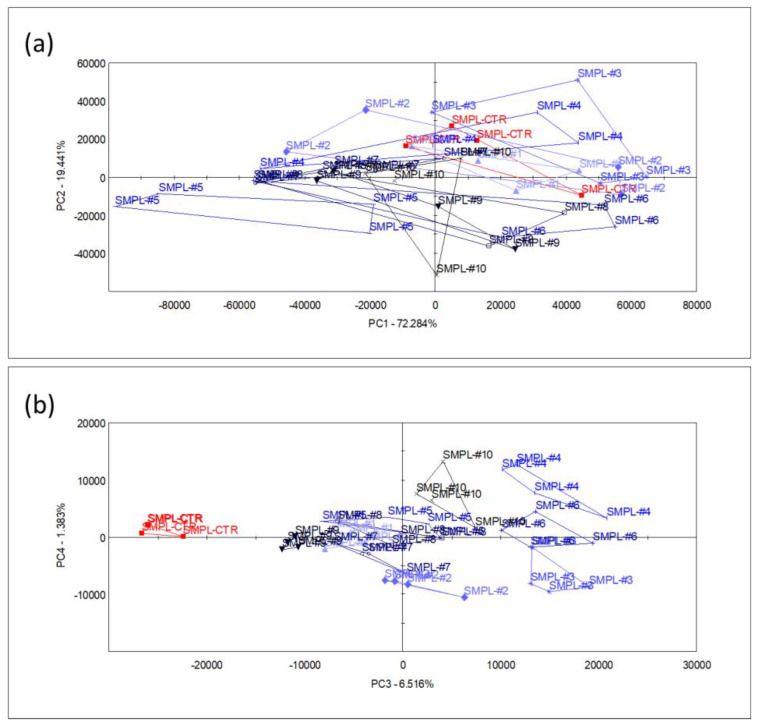
Score plots of PCA performed with data from all sensors, showing the planes of the 1st and 2nd principal components (**a**) and the 3rd and 4th principal components (**b**) (n = 4 replicates/sample, SMPL-CTR: milkshake with no supplementation (control); SMPL#1 to SMPL#10: milkshake samples with increasing levels of SP17-100 supplementation (0.2–2%).

**Figure 4 foods-11-03452-f004:**
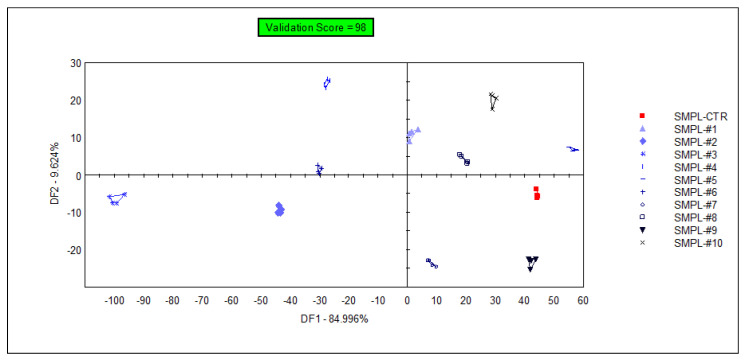
Graphical representation of the DFA classification of samples with different levels of S17-P100 supplementation obtained using data from all sensors (SMPL-CTR: milkshake with no supplementation (control); SMPL#1 to SMPL#10: milkshake samples with increasing levels of DHA supplementation (0.2–2%).

**Figure 5 foods-11-03452-f005:**
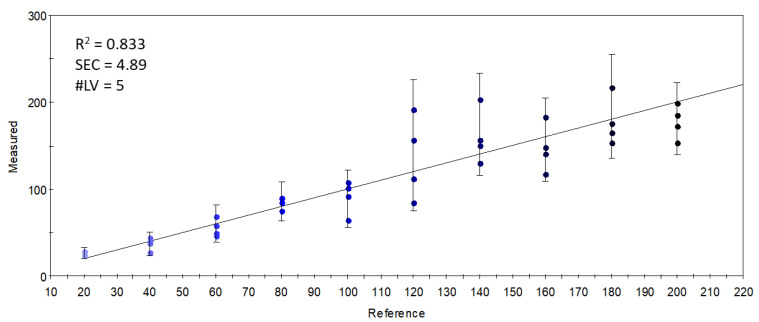
Y-fit of the PLSR calibration on the S17-P100 concentration (mg/in 10 g product) using data from all sensors (R^2^: determination coefficient; SEC: standard error of calibration; #LV: number of latent variables); the intensity of the colors from light blue to black along the Y-fit line represents the increase in concentration levels of the DHA supplementation.

**Figure 6 foods-11-03452-f006:**
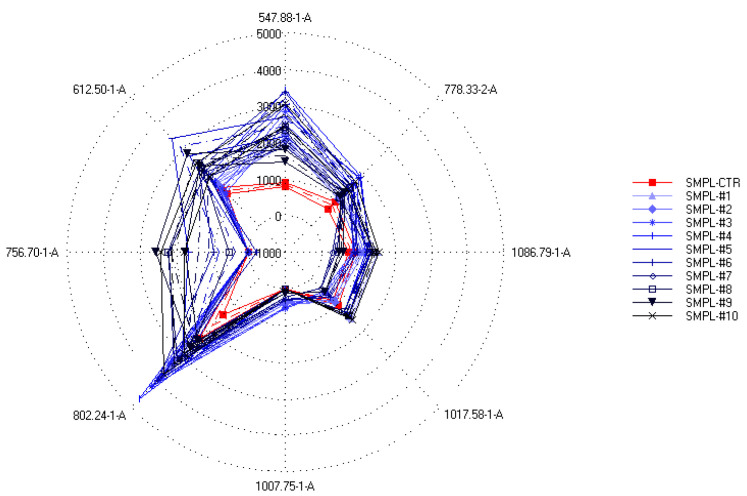
Spider plot of intensities (mean ± SD) of the sensors specifically selected to provide the best classification of the samples according to the different S17-P100 concentrations (SMPL-CTR: milkshake with no supplementation (control); SMPL#1 to SMPL#10: milkshakes with increasing levels of DHA supplementation (0.2–2%)).

**Figure 7 foods-11-03452-f007:**
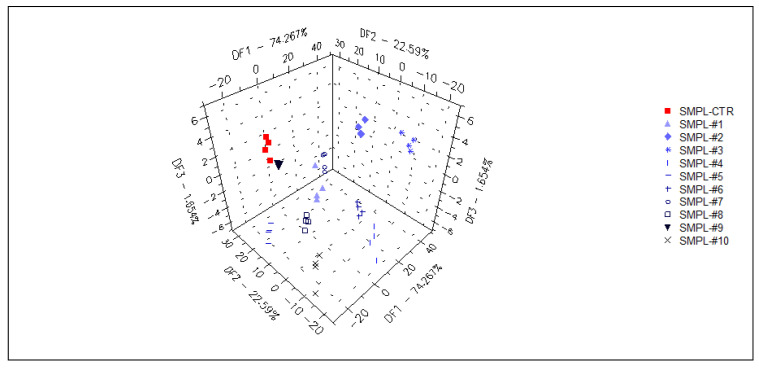
3D plot of the DFA classification of samples with different S17-P100 inclusion levels based on the data of the selected qualitative sensors (cross-validation score = 98%) SMPL-CTR: milkshake with no supplementation (control); SMPL#1 to SMPL#10: milkshakes with increasing levels of DHA supplementation (0.2%–2%).

**Figure 8 foods-11-03452-f008:**
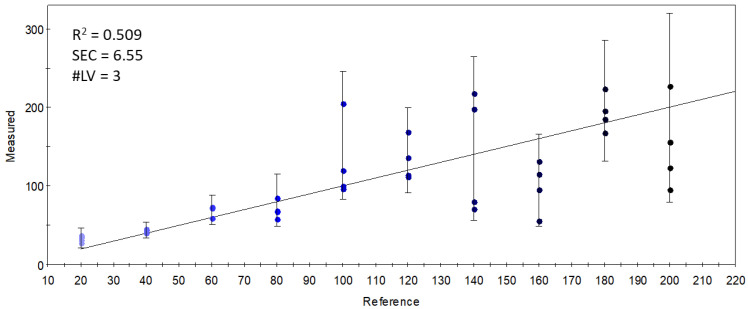
Y-fit of the PLSR calibration on the S17-P100 concentration (mg/10 g product) using data from the selected qualitative sensors (R^2^: determination coefficient; SEC: standard error of calibration; #LV: number of latent variables); the intensity of the colors from light blue to black along the Y-fit line represents the increase in concentration levels of the DHA supplementation.

**Figure 9 foods-11-03452-f009:**
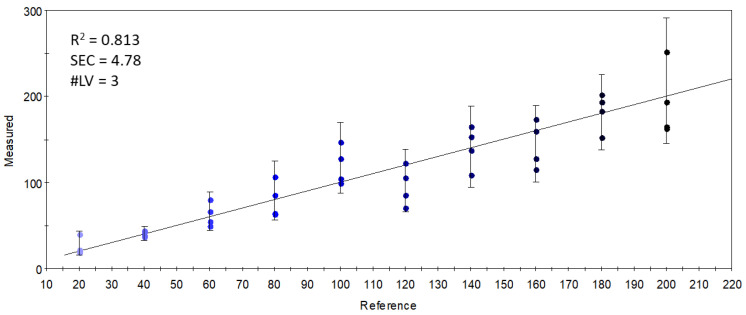
Y-fit of the PLSR calibration on the S17-P100 concentration (mg/10 g product) using data from the selected quantitative sensors (R^2^: determination coefficient; SEC: standard error of calibration; #LV: number of latent variables); the intensity of the colors from light blue to black along the Y-fit line represents the increase in concentration levels of the DHA supplementation.

**Table 1 foods-11-03452-t001:** Constituents of the commercial vanilla shake powder used in the experiment.

Ingredients	%
milk whey concentrate	69.2
hydrolyzed collagen	10
vegetable fat (coconut fat)	8
branched chain amino acids	4
freeze dried gelatin	4
Na-carboxy-methyl-cellulose	3
vitamin premix	1
vanilla aroma	0.3
beta-carotene	0.3
Sucralose	0.2

**Table 2 foods-11-03452-t002:** Quantitative and qualitative fatty acid composition of the S17-P100 food additive.

Fatty Acid	mg FA/g Sample	Weight % of FA
C4:0	0.24	0.05
C8:0	2.88	0.65
C10:0	3.38	0.77
C12:0	0.72	0.16
C14:0	24.2	5.50
C14:1n5	0.28	0.06
C15:0	1.81	0.41
C16:0	85.2	19.3
C16:1n7	0.61	0.14
C17:0	0.46	0.10
C18:0	5.60	1.27
C18:1n9c	48.1	10.9
C18:2n6c	5.72	1.30
C18:3n3	0.56	0.13
C20:0	0.74	0.17
C20:3n6	1.99	0.45
C20:4n6	3.19	0.72
C20:3n3	0.08	0.02
C22:0	0.72	0.16
C20:5n3 (EPA)	7.37	1.67
C24:0	0.68	0.15
C22:6n3 (DHA)	245.8	55.8
Total FA mg/g sample	440.3	
n3	253.8	57.6
n5	0.28	0.06
n6	10.9	2.48
n7	0.61	0.14
n9	48.1	10.9
n6/n3	0.04	
saturated	126.6	28.8
monounsaturated	49.0	11.1
polyunsaturated	264.7	60.1
EPA + DHA mg in 100 g	253.17	57.5

n3, omega-3; n5, omega-5; n6, omega-6; n7, omega-7; n9, omega-9; n6/n3, omega-6 to omega-3 ratio; EPA, eicosapentaenoic acid; DHA, docosahexaenoic acid.

**Table 3 foods-11-03452-t003:** Quantitative and qualitative fatty acid profiles of the original vanilla shake powder.

Fatty Acid	mg FA/g Sample	Weight % of FA
C4:0	0.02	0.02
C6:0	0.03	0.03
C8:0	2.86	3.07
C10:0	4.32	4.64
C11:0	0.03	0.03
C12:0	28.6	30.7
C13:0	0.04	0.04
C14:0	15.3	16.4
C14:1n5	0.23	0.25
C15:0	0.41	0.44
C16:0	17.2	18.5
C16:1n7	0.40	0.43
C17:0	0.21	0.23
C17:1n7	0.06	0.06
C18:0	5.99	6.44
C18:1n9t	0.05	0.05
C18:1n7t	0.03	0.03
C18:1n9c	12.1	13.0
C18:2n6t	0.07	0.08
C18:2n6c	4.12	4.43
CLA(9c,11t)	0.18	0.19
CLA(10t,12c)	0.02	0.02
C18:3n3	0.21	0.23
C20:0	0.14	0.15
C20:1n9	0.05	0.05
C20:2n6	0.01	0.01
C20:3n6	0.12	0.13
C20:4n6	0.09	0.10
C22:0	0.08	0.09
C20:5n-3 (EPA)	0.04	0.04
C24:0	0.06	0.06
Total FA mg/g sample	93.1	
n3	0.25	0.27
n5	0.23	0.25
n6	4.61	4.95
n7	0.49	0.53
n9	12.2	13.1
n6/n3	18.4	
saturated	75.3	80.9
monounsaturated	12.9	13.9
polyunsaturated	4.86	5.20

n3, omega-3; n5, omega-5; n6, omega-6; n7, omega-7; n9, omega-9; n6/n3, omega-6 to omega-3 ratio; EPA, eicosapentaenoic acid.

**Table 4 foods-11-03452-t004:** The graded food additive enrichment protocol (product composition).

No.	Shake Powder	S17-P100	Total Mass	Calc. DHA	Calc. EPA	Calc. EPA + DHA
	mg in 10 g					
1	9980	20	10,000	4.9	1.5	6.4
2	9960	40	10,000	9.8	3.0	12.8
3	9940	60	10,000	14.7	4.4	19.1
4	9920	80	10,000	19.6	5.9	25.5
5	9900	100	10,000	24.5	7.4	31.9
6	9880	120	10,000	29.4	8.9	38.3
7	9860	140	10,000	34.3	10.4	44.7
8	9840	160	10,000	39.2	11.8	51.0
9	9820	180	10,000	44.1	13.3	57.4
10	9800	200	10,000	49.0	14.8	63.8

S17-P100, brand name of the primarily docosahexaenoic acid enrichment used; EPA, eicosapentaenoic acid (C20:5 n3); DHA, docosahexaenoic acid (C22:6 n3).

**Table 5 foods-11-03452-t005:** Fatty acid profile (quantitative (mg/g product) and qualitative (%)) of the enriched milkshake sample series.

Sample No.	1		2		3		4		5	
Fatty Acid	mg/g	%	mg/g	%	mg/g	%	mg/g	%	mg/g	%
C4:0	0.02	0.02	0.02	0.02	0.02	0.02	0.03	0.03	0.02	0.02
C6:0		0.00	0.12	0.14	0.08	0.10	0.12	0.13	0.27	0.29
C8:0	1.07	1.36	3.20	3.72	2.92	3.48	3.28	3.85	3.95	4.22
C10:0	3.03	3.85	3.93	4.56	3.71	4.43	3.85	4.51	4.20	4.48
C11:0	0.02	0.03	0.02	0.02	0.03	0.03	0.03	0.03	0.02	0.02
C12:0	24.6	31.3	27.6	32.0	25.5	30.5	24.7	29.0	26.6	28.4
C13:0	0.03	0.04	0.03	0.04	0.04	0.05	0.05	0.06	0.05	0.05
C14:0	13.2	16.8	14.1	16.4	13.6	16.2	13.6	16.0	14.6	15.5
C14:1n5	0.18	0.22	0.15	0.17	0.15	0.17	0.18	0.21	0.20	0.22
C15:0	0.26	0.33	0.32	0.37	0.37	0.44	0.40	0.47	0.40	0.43
C16:0	16.2	20.5	16.6	19.3	16.9	20.2	17.5	20.5	17.4	18.5
C16:1n7	0.27	0.35	0.33	0.38	0.33	0.39	0.30	0.35	0.43	0.46
C17:0	0.14	0.17	0.17	0.20	0.19	0.23	0.21	0.24	0.21	0.22
C17:1n7	0.05	0.06	0.04	0.05	0.04	0.05	0.05	0.06	0.06	0.07
C18:0	5.28	6.71	5.30	6.15	5.65	6.74	5.66	6.64	5.94	6.34
C18:1n9t	0.02	0.02	0.05	0.05	0.04	0.05	0.04	0.04	0.03	0.04
C18:1n7t	0.01	0.01	0.01	0.01	0.02	0.02	0.02	0.02	0.01	0.01
C18:1n9c	10.1	12.8	9.82	11.4	9.41	11.2	10.0	11.8	12.9	13.7
C18:2n6t	0.04	0.06	0.06	0.07	0.05	0.06	0.05	0.06	0.09	0.09
C18:2n6c	3.24	4.11	3.23	3.75	3.14	3.74	3.40	3.99	4.03	4.31
CLA(9c, 11t)	0.12	0.15	0.13	0.15	0.15	0.18	0.13	0.15	0.15	0.16
CLA(10t, 12c)	0.02	0.02	0.01	0.01	0.01	0.01	0.01	0.01	0.01	0.01
C18:3n3	0.14	0.18	0.15	0.17	0.15	0.18	0.18	0.21	0.20	0.21
C20:0	0.09	0.11	0.11	0.13	0.11	0.13	0.11	0.13	0.13	0.13
C20:1n9	0.03	0.04	0.03	0.03	0.04	0.04	0.04	0.05	0.04	0.04
C20:2n6	0.01	0.01	0.01	0.01	0.01	0.01	0.00	0.00	0.01	0.01
C20:3n6	0.08	0.10	0.07	0.08	0.09	0.10	0.10	0.12	0.10	0.11
C20:4n6	0.06	0.08	0.08	0.09	0.07	0.08	0.08	0.09	0.10	0.10
C22:0	0.05	0.07	0.07	0.08	0.07	0.09	0.08	0.09	0.08	0.09
C20:5n3 (EPA)	0.01	0.01	0.02	0.02	0.03	0.04	0.05	0.05	0.06	0.06
C24:0	0.02	0.03	0.04	0.05	0.05	0.05	0.05	0.05	0.05	0.05
C22:5n3	0.07	0.08	0.08	0.09	0.07	0.08	0.09	0.10	0.15	0.16
C22:6n3 (DHA)	0.26	0.33	0.28	0.32	0.74	0.88	0.91	1.07	1.27	1.36
Total FA mg/g sample	78.7		86.1		83.8		85.3		93.7	
n3	0.48	0.61	0.52	0.61	1.00	1.19	1.23	1.44	1.68	1.79
n5	0.18	0.22	0.15	0.17	0.15	0.17	0.18	0.21	0.20	0.22
n6	3.56	4.53	3.58	4.16	3.51	4.19	3.78	4.43	4.50	4.80
n7	0.33	0.42	0.38	0.44	0.39	0.47	0.36	0.42	0.51	0.54
n9	10.1	12.9	9.9	11.5	9.5	11.3	10.1	11.9	12.9	13.8
n6/n3	7.44		6.84		3.52		3.08		2.68	
saturated	64.0	81.4	71.6	83.1	69.3	82.6	69.7	81.6	73.9	78.8
monounsaturated	10.6	13.5	10.4	12.1	10.0	12.0	10.7	12.5	13.7	14.6
polyunsaturated	4.04	5.13	4.10	4.76	4.51	5.38	5.00	5.86	6.18	6.59
EPA + DHA mg in 100 g	27.0	0.34	29.5	0.34	77.4	0.92	95.8	1.12	132.8	1.42
**Sample no.**	**6**		**7**		**8**		**9**		**10**	
**Fatty acid**	**mg/g**	**%**	**mg/g**	**%**	**mg/g**	**%**	**mg/g**	**%**	**mg/g**	**%**
C4:0	0.04	0.05	0.02	0.02	0.02	0.02	0.04	0.04	0.05	0.05
C6:0	0.40	0.44	0.04	0.05	0.07	0.08	0.37	0.37	0.40	0.41
C8:0	4.08	4.40	2.77	3.10	2.57	2.98	4.18	4.27	3.98	4.13
C10:0	4.13	4.44	3.82	4.28	3.43	3.99	4.26	4.35	4.06	4.21
C11:0	0.02	0.02	0.02	0.03	0.03	0.03	0.03	0.03	0.03	0.03
C12:0	25.3	27.2	25.1	28.0	21.8	25.4	26.4	27.0	25.2	26.2
C13:0	0.04	0.04	0.04	0.04	0.04	0.05	0.04	0.04	0.04	0.05
C14:0	13.9	15.0	14.0	15.7	12.5	14.5	14.7	15.0	13.9	14.4
C14:1n5	0.22	0.23	0.21	0.23	0.27	0.31	0.25	0.26	0.22	0.23
C15:0	0.41	0.45	0.42	0.47	0.43	0.50	0.43	0.44	0.44	0.46
C16:0	17.3	18.6	17.6	19.7	17.1	19.8	17.8	18.1	17.8	18.5
C16:1n7	0.46	0.50	0.40	0.45	0.49	0.57	0.50	0.51	0.52	0.54
C17:0	0.22	0.24	0.21	0.24	0.22	0.25	0.22	0.22	0.20	0.20
C17:1n7	0.07	0.07	0.07	0.07	0.07	0.08	0.08	0.08	0.07	0.08
C18:0	5.85	6.30	6.02	6.74	5.68	6.60	5.69	5.81	5.57	5.78
C18:1n9t	0.05	0.05	0.04	0.05	0.03	0.04	0.05	0.05	0.05	0.05
C18:1n7t	0.02	0.03	0.02	0.02	0.02	0.03	0.01	0.01	0.02	0.02
C18:1n9c	13.3	14.3	11.6	13.0	13.2	15.4	14.1	14.4	13.9	14.4
C18:2n6t	0.09	0.10	0.08	0.09	0.09	0.10	0.07	0.08	0.08	0.08
C18:2n6c	4.14	4.46	3.77	4.22	3.94	4.57	4.34	4.43	4.08	4.23
CLA(9c, 11t)	0.17	0.19	0.16	0.18	0.20	0.23	0.19	0.20	0.24	0.24
CLA(10t, 12c)	0.01	0.01	0.01	0.02	0.01	0.01	0.01	0.01	0.01	0.01
C18:3n3	0.21	0.22	0.19	0.21	0.24	0.27	0.24	0.24	0.24	0.25
C20:0	0.14	0.15	0.13	0.15	0.13	0.15	0.12	0.12	0.13	0.14
C20:1n9	0.05	0.05	0.04	0.05	0.05	0.06	0.06	0.06	0.06	0.06
C20:2n6	0.01	0.01	0.01	0.01	0.02	0.02	0.01	0.01	0.01	0.02
C20:3n6	0.14	0.15	0.13	0.15	0.14	0.17	0.16	0.16	0.16	0.17
C20:4n6	0.11	0.11	0.11	0.13	0.13	0.15	0.14	0.14	0.16	0.16
C22:0	0.10	0.11	0.09	0.10	0.08	0.10	0.09	0.09	0.10	0.10
C20:5n3 (EPA)	0.07	0.08	0.08	0.08	0.12	0.14	0.13	0.13	0.16	0.16
C24:0	0.05	0.06	0.05	0.06	0.06	0.07	0.07	0.07	0.07	0.07
C22:5n3	0.16	0.17	0.14	0.16	0.18	0.21	0.21	0.22	0.24	0.24
C22:6n3 (DHA)	1.72	1.85	1.92	2.14	2.70	3.13	2.92	2.99	4.12	4.28
Total FA mg/g sample	92.9		89.3		86.1		97.9		96.4	
n3	2.16	2.32	2.32	2.60	3.23	3.75	3.50	3.57	4.76	4.93
n5	0.22	0.23	0.21	0.23	0.27	0.31	0.25	0.26	0.22	0.23
n6	4.67	5.02	4.29	4.80	4.52	5.25	4.93	5.04	4.74	4.92
n7	0.55	0.60	0.48	0.54	0.58	0.68	0.60	0.61	0.60	0.63
n9	13.4	14.4	11.7	13.1	13.3	15.5	14.2	14.5	14.0	14.5
n6/n3	2.16		1.85		1.40		1.41		1.00	
saturated	71.9	77.4	70.4	78.8	64.2	74.5	74.4	76.0	72.1	74.8
monounsaturated	14.2	15.3	12.4	13.8	14.2	16.5	15.1	15.4	14.8	15.4
polyunsaturated	6.82	7.34	6.61	7.40	7.75	9.01	8.43	8.61	9.49	9.85
EPA + DHA mg in 100 g	179.1	1.93	199.2	2.23	281.3	3.27	305.0	3.12	428.0	4.44

n3, omega-3; n5, omega-5; n6, omega-6; n7, omega-7; n9, omega-9; n6/n3, omega-6 to omega-3 ratio; EPA, eicosapentaenoic acid; DHA, docosahexaenoic acid.

**Table 6 foods-11-03452-t006:** Assignment of possible volatile compounds to the retention indices of the selected qualitative sensors using the AroChemBase database (Alpha MOS, Toulouse, France).

Retention Index	Compound 1 *	Compound 2 *	Compound 3 *
547-1-A	tert-buthylmethylether	1-propanol	2-propanol
612-1-A	ethylacetate	acetic acid	–
756-1-A	ethyl isobutyrate	Pyrrole	–
802-1-A	ethyl butyrate	propyl propanoate	–
996-1-A	ethyl hexanoate	butyl butanoate	–
1017-1-A	alpha-terpinene	1,4-cinelole	acetylpyrazine
1086-1-A	pentyl butanoate	benzyl butanoate	–
778-2-A	pentanal	propyl acetate	pentan-2-one

* Compounds 1, 2, and 3, represent the likelihood of the identified compound being associated with the identified retention indexes (Compound 1: highest possibility of occurrence, Compound 2: the next highest to occur, and Compound 3: the lowest to occur), and “–“means no compound 3 was identified at that specific intensity.

## Data Availability

The data are available from the corresponding author.

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
