# Peer review of "Micro-Encapsulated Microalgae Oil Supplementation Has No Systematic Effect on the Odor of Vanilla Shake-Test of an Electronic Nose"

_foods, 2022, doi:10.3390/foods11213452_

Round 1
Reviewer 1 Report
The manuscript under appreciation is about the fortification of a milk-based vanilla shake powder with micro-encapsulated microalgae oil.
Generally, the manuscript is interesting and provides novelty.
The following comments are to be taken account by the authors:
In the “Introduction” section, it would be useful to emphasize in the novelty of this study, and add information about other products that have been possibly fortified with micro-encapsulated microalgae oil. Is this study the first attempt of micro-encapsulated microalgae oil fortification?
In the “Materials and Methods” section, line 98: please be more specific about the use of internal standard, e.g. the quantity added and how you used it to calculate the concentration of FA? How many replicates did you performed regarding the FA analysis?
Regarding the “Discussion” section, do the authors believe that the organoleptic characteristics may have been altered? Would this enriched product be acceptable to consumers?
Reviewer 2 Report
This work aims to ascertain the efficacy of an added n3 FA fortification brand (micro-encapsulated microalgae oil) in fulfilling dietary recommendations for n3 FA-enriched food products and to test the application of the e-nose in profiling the odor of the n3 FA-enriched vanilla milkshakes. The basic hypothesis to be tested was if the characteristically fish odor-providing oil source has a detectable systematic distorting effect on the odor profile of a vanilla shake product.
Despite the interesting application of the work and using e-nose, the way that the work is presented does not suit. There are lots of issues with this article, and I will try to enumerate some.
The novelty of the work must be stated as expected for an article in Foods.
The title is too long and confusing.
There is no clear statement of a scientific problem being addressed, which limits the significance of the work considerably. The authors purchase the vanilla shake powder and micro-encapsulated, marketed algae oil-based product with the brand name S17-P100 and the explanation of how and why these are used must be better framed.
Written English must be revised thoroughly in the entire manuscript; there are lots of confusing sentences that make the article extremely difficult to read.
Statistical analysis should be provided on the figures and tables. Some points to consider below
Line 26: in DHA (and n3 fatty acid) incorrect sentence
The abstract must be concise and provide important results. Please rewrite the abstract.
Round 2
Reviewer 2 Report
The revised version is slightly improved in the section 3.3. microbiology more details should be added. Also in the discussion section the microbiology data is not well discussed
Why did the authors not make storage periods for milkshakes? to follow the oxidation of oil and change in the odor
The MS still need lot of work.
